# A measurement strategy to address disparities across household energy burdens

Eric Scheier [1,2] & Noah Kittner [2,3,4 ✉]

Energy inequity is an issue of increasing urgency. Few policy-relevant datasets evaluate the energy burden of typical American households. Here, we develop a framework using Net Energy Analysis and household socioeconomic data to measure systematic energy inequity among critical groups that need policy attention. We find substantial instances of energy poverty in the United States – 16% of households experience energy poverty as presently defined as spending more than 6% of household income on energy expenditures. More than 5.2 million households above the Federal Poverty Line face energy poverty, disproportionately burdening Black, Hispanic, and Native American communities. For solar, wind, and energy efficiency to address socioeconomic mobility, programs must reduce energy expenditures by expanding eligibility requirements for support and access to improved conservation measures, efficiency upgrades, and distributed renewables. We recommend the United States develop a more inclusive federal energy poverty categorization that increases assistance for household energy costs.

---

[1] Emergi Foundation, Carrboro, NC, USA. [2] Environment, Ecology, and Energy Program, The University of North Carolina at Chapel Hill, Chapel Hill, NC, USA. [3] Department of Environmental Sciences and Engineering, Gillings School of Global Public Health, The University of North Carolina at Chapel Hill, Chapel Hill, NC, USA. [4] Department of City and Regional Planning, The University of North Carolina at Chapel Hill, Chapel Hill, NC, USA. ✉email: kittner@unc.edu

H ousehold energy use for services such as cooking[1] or space heating and cooling[2] is crucial for decent living conditions[3]. Unaffordable energy is a persistent trend[4] that is negatively related to social cohesion, climate change responses, and disproportionate environmental impacts on low-income populations and minority groups[5]. Furthermore, energy inequity is not just a lack of money to meet basic energy needs—it is a lack of access to the capabilities[6] that enable a sustainable and prosperous society built on just principles[7]. Energy inequity could have significant implications for navigating sustainable development and meeting societal goals around decarbonization and energy use. In this study, we demonstrate the magnitude of energy inequity in the United States (US) using a metric informed by Net Energy Analysis (NEA). Without a set of inclusive indicators and data tools to examine energy inequity, many households that are at risk of energy poverty and injustice may remain unidentified. This analysis applies lessons from NEA to address energy poverty in the US.

Many frameworks are currently being explored to understand energy poverty and equity, while differentiating between related concepts. In a thematic exploration of energy equity, Brown et al. identify energy access, energy poverty, energy insecurity, and energy burden as key concepts for understanding the issue[8], but quantitative measurement of these concepts has been limited. Pachauri & Rao establish measures for the sustainable development context that incorporate periods when energy is available, the quality of voltage supplied, the reliability in terms of the number of disruptions, the capacity in terms of power available, the consumption levels allowed per day, and affordability of the standard consumption package as a percentage of household income[9]. Energy metrics have been assessed quantitatively across several countries. Though some aspects, such as formal disconnections from energy service[10], are translatable to the US context, these methods require normalizing many variables amongst different types of data and are overly broad for applications in areas where electricity access is relatively high and reliable.

Of such areas, the United Kingdom (UK) has a richer history of incorporating energy poverty formally into its government programs: since 2000, the UK has used some form of an energy burden metric to assess whether households are facing energy poverty and determine the level of support that they require as a result[11]. This metric has evolved from a simple ratio of household income and energy expenditures to one that incorporates building efficiency ratings and average incomes in the community. While the European Union (EU) currently lacks a unified metric for energy poverty, similar metrics have been developed in member countries, such as a metric which compares incomes and expenditures to local averages and absolute heating needs to determine energy poverty levels in Italy[12] or a multidimensional index of building quality and ability to pay bills in Poland[13].

Prior research of household prosperity in limited global jurisdictions using these sorts of measures has found that gender, age, housing age[14], tenure type[15], energy inefficiency[16], education, employment[17], geography[18], socioeconomic status[19], race/ethnicity[20], and macroeconomic conditions[21] are associated with high energy burdens in various geographical areas. The US lags in part due to a lack of metrics and tracking, and this paper develops a tool to show how net energy is a valuable resource to evaluate energy burden and inequality.

In the US, energy inequity is a significant challenge as families struggle to meet monthly bills and live paycheck to paycheck[11]. There is a growing disparity between wealthier and lower-income households based on their abilities to meet basic energy needs[8]. While per-unit residential energy prices have increased below the rate of inflation in the US since the 1980s[22], many households still struggle to make utility bill payments and are especially vulnerable to economic shocks[21].

Ross and Drehobl performed distinct urban[16] and rural[14] analyses to describe energy inequity in the US. While limited by geographic and demographic focus and a lack of peer-review, these studies have established the proportion of income ($G$) spent on energy expenditures ($S$), or energy burden ($E_b$), as the standard benchmark for energy poverty in the US today (Eq. 1). The US Department of Energy (DOE) significantly improved upon Ross and Drehobl's underlying methodology by assembling its Low-Income Energy Affordability Dataset (LEAD)[23], which estimates incomes and energy expenditures for most households in the US at a census-tract scale.

$$\text{Energy Burden} \, (E_b) = \frac{S}{G} \qquad (1)$$

NEA offers potential support to understanding energy equity through the use of formally defined Energy Return Ratios (ERRs) like $E_b$ that articulate the relationships among energy flows within complex systems[24]. Net Energy Return (NER), which describes the newly released potential to do work as a result of some activity, is recommended as a basis for future analysis[25], especially in the study of macro-energy systems like the US residential housing stock[26].

Here, we examine the relationship between energy spending and household energy income and observe spatial disparities across a census-tract scale—particularly across race and ethnicity, income, housing tenure, and educational attainment. A difference of $500 in annual energy expenditures dramatically improves many middle and high-income households' ability to enjoy benefits from energy services for low cost (2–5% of annual income), but basic energy expenditures comprise approximately 14% of annual earnings for low-income households. The categorizations for federal assistance programs that are based on poverty line indices fail to capture at least 5.3 million households that would benefit from energy support and assistance. Households in communities of color experience energy poverty at a rate 60% greater than those in white communities based on this study. Additionally, state-level analysis demonstrates wider spatial disparities across states such as Maine, where residential heating oil and fuelwood remain common heat energy sources and states such as Alabama and Mississippi which have among the lowest net energy returns on average. This empirical analysis fills a gap in the current discussion about energy equity by providing a framework to evaluate disparities and include more households in energy poverty metrics that are aligned with the assessment of energy flows through biological, physical, and economic systems.

## Results

**Applying net energy return to energy equity.** The main contribution of the study is the application of NER as an indicator of household energy poverty by utilizing a set of energy burden metrics based on NER to represent the disparity across those who face significant energy burdens in the US with those who do not. NER displays critical thresholds where more households face energy and income challenges.

The NER of a process is a relationship between the gross amount of energy extracted and the amount of embodied energy directed toward extraction[27]. For households extracting income from the economy, these ratios can be composed of gross income ($G$) and spending on energy ($S$).

Household NER ($N_h$) represents the net earnings a household receives for every expenditure on secondary energy, defined according to Eq. (2):

$$\text{Household Net Energy Return} \, (N_h) = \frac{G - S}{S} \qquad (2)$$

The NER is the standard metric of NEA because it reflects the role of energy as an input to a value-generating process in ecosystems[28].

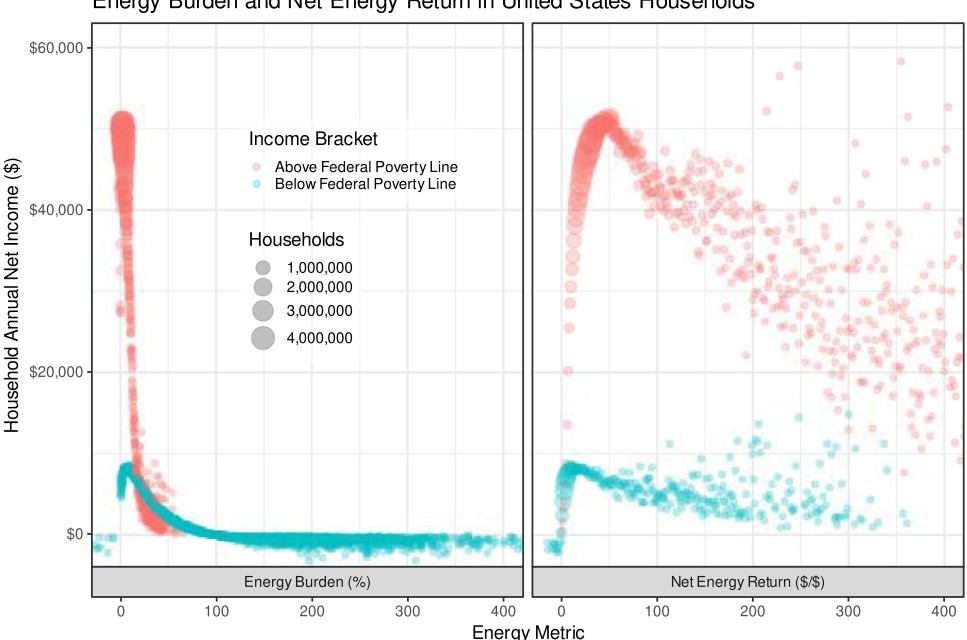

**Fig. 1 Display of the relationship between $E_b$ and $N_h$ with net income (gross income - energy expenditures) for US households.** While $E_b$ appears inversely correlated to net income, this is primarily driven by a long tail of households with zero or very low incomes, often with energy expenditures exceeding their incomes. Due to the structure of the $E_b$ equation (Eq. 1), the $E_b$ of these households approaches infinity and cannot be captured on the standard 0–100% scale the metric is intended to be interpreted within. Since LEAD is estimated and provided for the express purpose of exploring low-income communities, we are hesitant to discard these households as outliers. $N_h$ frames the same dataset on a scale without the long tail. $N_h$ appears positively related to income, and most communities appear within a few orders of magnitude. Utilizing $N_h$ avoids discarding low-income communities as outliers in energy poverty analysis. Furthermore, $N_h$ offers a way to view the relationship between energy expenditures and income such that the wide disparity between those in broader poverty is immediately apparent: many households with moderate-to-high $E_b$s are actually higher-income households with high energy expenditures, making their $N_h$s quite high (e.g., >$100 of income per $1 of energy spending), which is only visible on an $N_h$ scale.

$E_b$ is the metric of choice in the energy insecurity literature due to its presumed interpretability as a percentage[14,16,19,21].

While most ERRs, including NER, are hyperbolic paraboloids, NER has several useful mathematical properties: it can smoothly handle systems with negative incomes and energy costs, accept households with zero income, and emphasize extreme incomes and energy costs in an interpretable fashion, as shown in Fig. 1. While $E_b$ appears inversely correlated to income, this is primarily driven by a long tail of households with zero or very low incomes, often with energy expenditures exceeding their incomes (approximately $n = 118{,}000$ households in the dataset have $E_b > 100\%$ or $E_b < 0\%$). Due to the structure of the $E_b$ equation (Eq. 1), the $E_b$ of these households approaches infinity and cannot be captured on the standard 0-100% scale the metric is intended to be interpreted within: around 37,000 homes have an infinite energy burden. Since LEAD is estimated and provided for the express purpose of exploring low-income communities, we are hesitant to discard these households as outliers. $N_h$ provides a framing of the same dataset that allows for exploration of most households on a similar scale without the long tail. $N_h$ appears positively related to income, and most communities appear within a few orders of magnitude. Utilizing $N_h$ avoids discarding low-income communities as outliers in energy poverty analysis. Furthermore, $N_h$ offers a way to view the relationship between energy expenditures and income such that the wide disparity between those in broader poverty is immediately apparent. Many households with moderate-to-high $E_b$s are actually higher-income households with high energy expenditures, making their $N_h$s quite high (e.g., >$100 of income per $1 of energy spending). Almost 100% of the households in the dataset have net incomes between −$1000 and $60,000, with $N_h$s (or $E_b$s) of between −10

(−10%) and 400 (400%). Only households with no energy costs are excluded from the analysis, whereas households with no energy costs or incomes must be excluded from an analysis utilizing $E_b$.

For the discussion of household energy poverty, we are primarily interested in how households of different characteristics are distributed according to their $N_h$s, representing how many net dollars are earned by a household for every dollar it spends on energy. Since the $N_h$ is unitless but is a ratio of return on investment, we present it below interchangeably with no units or in units of $/$ depending on context. Other proposed indicators of energy poverty may be similarly examined in this manner.

**Application to energy poverty.** While a variety of thresholds have been developed and explored, energy-poor households in the US are commonly defined in terms of $E_b$ as those with an expenditure of greater than 6% of household income on energy based on the logic that energy expenditures should not be greater than 20% of housing expenses, which themselves should not exceed 30% of household income[8]. Calibrating our $N_h$ analysis to this level will help gauge different thresholds of energy poverty and benchmark the results of this paper to the energy poverty literature while acknowledging the continuum of experiences across household energy consumption. Translated into its relative level for $N_h$, the energy poverty line $N_h$ is defined according to Eq. (3) as approximately 16.

$$E_B^* = \frac{S}{G} = 6\%$$
$$N_h^* = \frac{G-S}{S} \text{ such that } \frac{S}{G} = 6\% \qquad (3)$$
$$N_h^* \approx 16 \Rightarrow \text{Household at Energy Poverty Line}$$

This means that a household that earns less than approximately $16 of income for every dollar it spends on secondary energy will be considered to be in energy poverty by the traditional $E_b$ accounting method. An $N_h$ of approximately 16 or lower is equivalent to an $E_b$ of 6% or higher. This threshold is arbitrary and may not be suitable in situations where households fall very close to this line or where the numbers of family members or measures of certain building characteristics vary widely. Simply, it is presented as a benchmark. We examine the $N_h$ at a community scale across the US in Table 1.

**The US household net energy landscape.** We use LEAD[23] ($n$ = 113.2 million households) to evaluate the $N_h$ dynamics across the

US. The average US home in the dataset has an income of $41,922 and an annual energy expenditure of $1219 ($102/month), which equates to an average $E_b$ of 3% or an average $N_h$ of 33.4. While the dataset slightly overrepresents households below the median income due to the limitations of the statistical methods used to compile them, it represents estimates for approximately 94–95% of the 119.5–120.0 million family-occupied households evaluated by the American Community Survey. Table 1 shows a summary of these data for households and their average statistics delineated by their incomes relative to the median income of similar size families in the same metropolitan area or non-metropolitan county, known as Area Median Income (AMI). We can see that high burdens are found mostly in the very-low-income group, with an average $E_b$ of 14% or $N_h$ of 6.2. Income drives the escape from energy poverty: middle and high-income groups do not spend drastically different amounts on energy (21% and 63% more, respectively) but earn 3 and 10 times that of the very low-income group, respectively. An $E_b$ of 14% means that $71 per month is spent on energy, which is quite high on a monthly income of $510. The additional $15 per month that the next rung of moderate-income groups spends on energy represents a minuscule proportion of their income (1%).

Another contribution of this work is that by using $N_h$, we can display these data spatially across the US to explore how different communities are experiencing energy outcomes as in Fig. 2 and investigate specific communities at multiple scales such as census tract, county, state, and regional as in Fig. 3. Furthermore, we can break down the data among meaningful subsets as in Fig. 4 and examine state-by-state trends as in Fig. 5. This is useful because the presence of energy inequity is harder to see in urban areas compared to rural areas for which census tracts are a larger physical area or when certain household characteristics such as primary heating fuel are related to widely differing energy outcomes in the same area. Additionally, it shows a spatial variation of energy poverty that includes 5.3 million more households that would not be captured by traditional poverty metrics because their incomes are too low ($E_b$ > 100%) or too high (above the FPL). By processing disparate data sources into a

**Table 1 Average annual household energy expenditures, incomes, net incomes, $N_h$s, and $E_b$s portrayed for different income groups based on their relationship to AMI.** Note that the statistics for $E_b$ and $N_h$ are calculated per cohort after averaging the income and expenditure statistics to avoid the effect of extreme values skewing the interpretability of these metrics as described in the text. For example, the actual weighted average $E_b$ for households from 0-30% AMI is 16,451%, and the average $N_h$ for this cohort is 8.6 due to subsets of the dataset with very low incomes or low energy expenditures. $N_h$ is not as susceptible to this skewing effect.

| Metric name | 0-30% AMI | 30-80% AMI | Above 80% AMI |
|---|---|---|---|
| Households in sample | 15.9 million | 29.7 million | 67.6 million |
| Annual income (G) | $6120 | $20,671 | $59,689 |
| Annual energy expenditures (S) | $853 | $1034 | $1386 |
| $E_b$ (S/G) | 14% | 5% | 2% |
| Net income (G-S) | $5267 | $19,637 | $58,303 |
| $N_h$ ([G-S]/S) | 6.2 | 19 | 42.1 |

AMI Area Median Income, $N_h$ Household Net Energy Return, $E_b$ Household Energy Burden

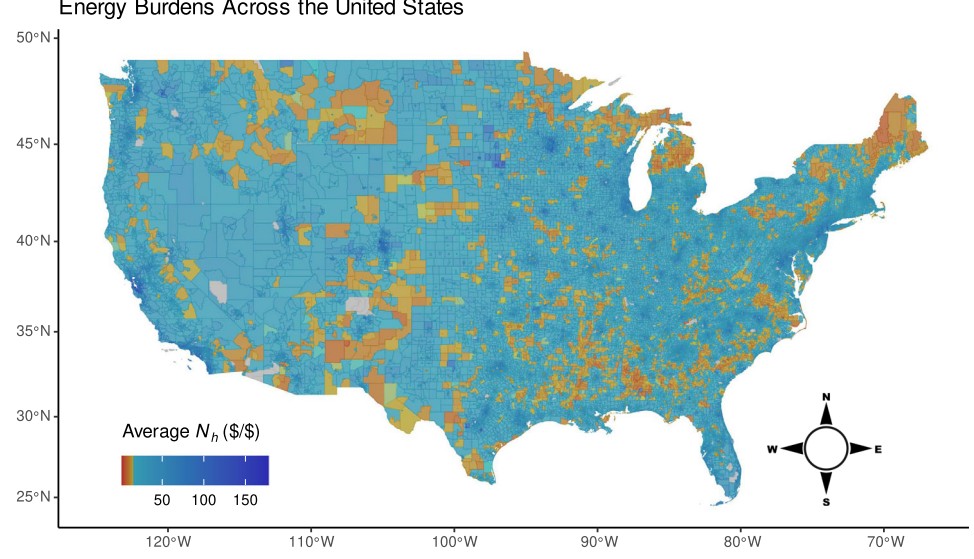

## Energy Burdens Across the United States

Average $N_h$ ($/$)

50   100   150

$N_h$ : Household Net Energy Return

**Fig. 2 Map of the average net earned income per secondary energy expenditure for each census tract in the continental US.** Shades of yellow and red, respectively, indicate communities at or below the energy poverty line as defined by earning approximately 16 dollars or less in income per dollar of energy expenditures. This corresponds to the traditional definition of energy poverty as spending 6% or more of income on energy. Low $N_h$s can be starkly observed in the Black Belt across the Southeastern US, Hispanic communities near the US-Mexico border, Native American lands, and rural New England.

coherent structure and providing a convenient open-source tool for others to do the same, these data can be used in urban planning, public policy, and other relevant contexts.

Displayed geospatially in Fig. 2, the Black Belt in the American Southeast is visibly perceptible as an area of high burden, indicating that low-$N_h$ follows racial lines. Likewise, border populations and immigrant populated areas in the Southwest have higher burdens, as do Native American lands. High burdens

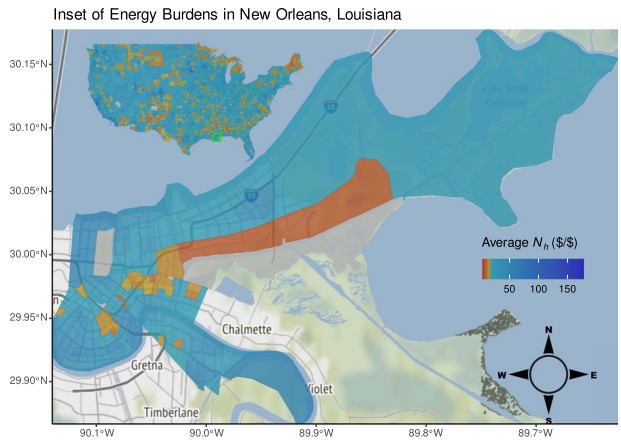

**Fig. 3 An inset focused on the city of New Orleans, Louisiana (Orleans Parish).** This view shows an example of the dynamics of energy poverty in an urban area where the presence of energy inequity is harder to see compared to rural areas for which census tracts are a larger physical area.

can also be seen in rural Northeast states where heating burdens are high. $N_h$ allows a nuanced view of these widely ranging income dynamics by portraying them on a scale that matches the scope of the issues: areas of high energy burden (close to $N_h = 0$) are visible in orange while highly affluent areas are also visually perceptible as dark blue even though these groups' average metrics are multiple orders of magnitude apart.

Urban inequity results in lower $N_h$ populations not showing up in many dense or gentrified urban areas such as the San Francisco Bay Area, New York City, and New Orleans, as shown in Fig. 3. The pervasiveness of urban energy poverty in Detroit has been studied extensively and shown to have distinct geographic boundaries down to the street level[19]. While some of these conclusions are supported by existing evidence and literature, they should be confirmed with a rigorous analysis of this dataset using the $N_h$ metric for the reasons explained in Applying NER to Energy Equity: extremely low-income households are not visible on a 0–100% scale, and households with no income are often discarded as outliers. Additional analyses should incorporate additional demographic and household dimensions due to potential disparities within census tracts since diverse neighborhoods may not be represented accurately by aggregate census-tract metrics.

From this high level, we can see in Fig. 4a that approximately 16% of households in the US experience energy poverty. Displaying these communities defined by their relationship to the US Federal Poverty Line (FPL), which indicates income poverty status according to government policy, in Fig. 4b provides a stark picture. While 94% of households below the FPL also face energy poverty, more than 5.2 million of those households above the FPL face this scarcity, underscoring the relative burden of

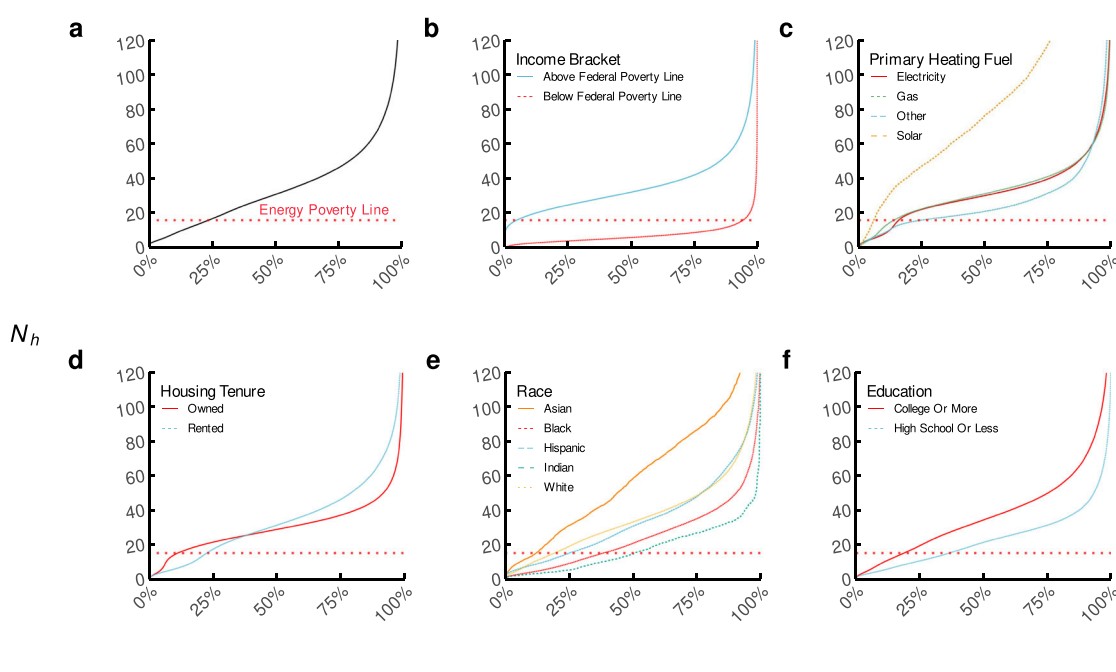

**Fig. 4 The distribution of $N_h$s across different household characteristics.** Subfigure (**a**) shows the overall distribution. Subfigure (**b**) shows the difference among those groups of households above and below the Federal Poverty Line. Subfigure (**c**) shows the difference among groups of households identified by their primary heating fuel. Subfigure (**d**) shows the difference based on whether they are renters or owners. Subfigure (**e**) shows the difference based on the most prominent race in the census tract of each cohort. Subfigure (**f**) shows the difference based on the most prominent education history in the census tract of each cohort.

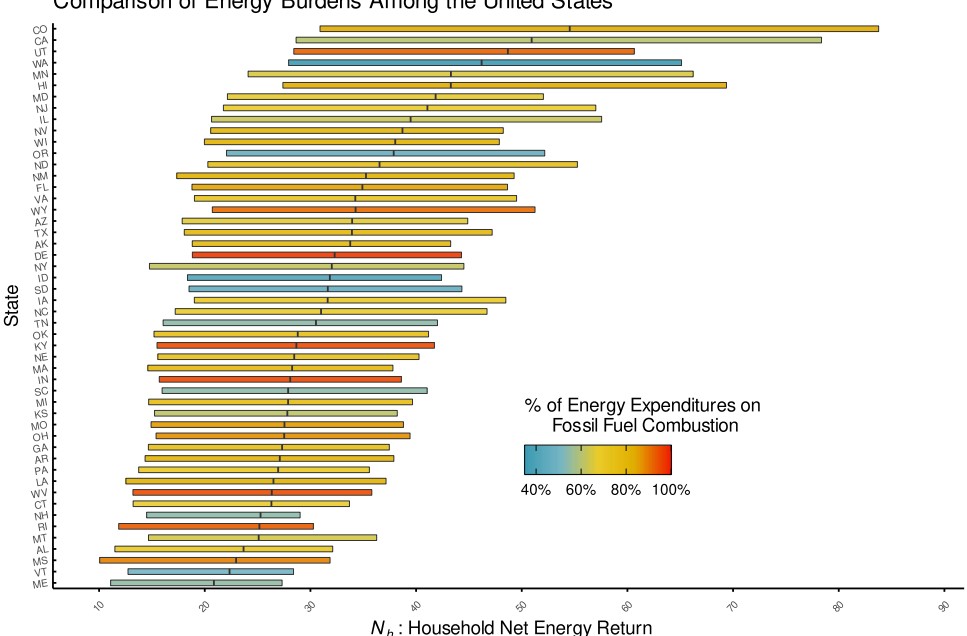

**Fig. 5 A comparison of $N_h$s among each state in the US.** The bars are sorted by median household $N_h$ and represent the interquartile range (25%-75% percentiles) of household $N_h$s colored by the percent of household energy expenditures in each state that goes to support fossil-fuel combustion, whether directly through natural gas purchases or indirectly through the electricity grid in each state. Natural gas purchases are assumed to be entirely combusted by the end-user, and electricity purchases are divided into their respective sources according to the US Environmental Protection Agency's (EPA) Emissions and Generation Resource Integrated Database (eGRID) for each state. Other expenditures are divided according to the primary heating fuels other than electricity or natural gas used by American households according to the Census, which are approximately 75% fossil-fuel combustion. The figure suggests that a reliance on fossil-fuel combustion does not lead to a more affordable energy system for end-users.

energy expenditures as a poverty trap and the inadequacy of FPL as an indicator of energy poverty in particular. When we break the group of relatively prosperous households into subsets as outlined in Table 1, we find that 32% of households living at 30–80% of their AMI are experiencing energy poverty. This suggests that energy poverty may be a useful metric for identifying households at risk of other forms of poverty. However, we find that most households experiencing energy poverty are also suffering from a broader lack of access to resources characterized by income-based poverty. Given that the quality of energy used by low-income households is expected to be of similar inherent usefulness, this stark contrast in $N_h$ on the households' energy investments is surprising.

Figure 4c shows that households with solar as a primary heating fuel have a higher $N_h$ ($N_h = 111$) than those which rely on other fuel sources ($N_h = 33$), even for households far below the income poverty line: the average $N_h$ for energy-impoverished households utilizing solar power is 19, compared to 7 for those relying on any other fuel source. The slope of the $N_h$ density lines in Fig. 4c indicates that these lower-income households seem to experience a different rate of return on $N_h$ from the benefits of increased energy adoption than higher-income households. Why are certain households not receiving the same benefits of their fuel source across the distribution of incomes? This could be due to lower-income households' low consumption, meaning that the potential savings from implementing energy efficiency measures are lower than for high-income households according to the prebound effect identified by Sunikka-Blank and Galvin[29] and Cong et al.[30]. Despite technological progress and rapid declines in technology costs and availability for cleaner, renewable electricity generation options such as solar photovoltaics and energy storage, many households cannot take advantage. Most households do not have access to advancements such as low-cost rooftop solar or energy efficiency upgrades that improve air quality, lower

greenhouse gas emissions, and save money[31]: only 18% of households that have adopted rooftop solar have been below the median household income in the US[32].

Examining these dynamics by the status of homeownership in Fig. 4d reveals further disparities. Though renters and homeowners are similarly distributed below the energy poverty line (28% of renters and 17% of homeowners are in energy poverty), there appears to be an advantage of homeownership from a net energy perspective for lower-income households ($N_h = 37$ for homeowners versus $N_h = 39$ for renters). Only at a relatively high $N_h$ do renters seem to have an advantage: owners of multi-family apartments earn 2.4 times as much as renters of single-family homes when normalized by energy expenditures. Renters face systemic disadvantages in the energy transition; they typically pay the home's energy costs while the landlord controls infrastructure upgrades, commonly understood as the split incentive problem[33]. Tenure matters for more than just equity itself: renters are less likely to take actions to improve their $N_h$s due to a lack of property rights and split incentives. Even when action is taken to improve the energy efficiency of a rental building, tenants are less likely to see any economic benefits from it.

Furthermore, Fig. 4e shows that $N_h$ varies widely by racial demography. Asian households have the highest $N_h$ across the entire population distribution ($N_h = 65$), and Indian households have the lowest ($N_h = 18$), with Black households a close second from the lowest ($N_h = 26$). These relative positions are the same across the entire population distribution, with only White ($N_h = 38$) and Hispanic ($N_h = 36$) populations showing different relative $N_h$s across the population. Households in communities of color experience energy poverty at a rate 60% greater than those in white communities. Education level also seems to be correlated with disparate $N_h$ outcomes according to Fig. 4f, with a wide gap between those households in areas with mostly high-school ($N_h = 25$) or college-educated ($N_h = 40$) populations.

Assessing the $N_h$s among different states in Fig. 5 presents a counterintuitive picture of how states address energy poverty and energy equity. $N_h$ can equate communities that experience high energy costs and low incomes with those with high incomes and even higher energy costs. This explains why states such as Connecticut ($N_h = 26$) and Vermont ($N_h = 23$), where 47% and 30% higher than average electricity prices may pose affordability threats for communities affected by higher prices, are similarly positioned on the list to states such as Mississippi ($N_h = 22$) and Alabama ($N_h = 24$), which have significant low-income populations and low per-unit energy prices (22% and 8% lower than average, respectively). Not only are households in these states falling behind in terms of income, but net incomes are lower relative to energy expenditures than neighboring states and other parts of the country. This may be appropriate: while the equity issues in Southeastern states are well studied, states such as Maine that continue to utilize residential heating energy sources like oil and fuelwood may suffer not only from lack of efficiency but also health impacts. States may need to pay attention to these dynamics from an affordability perspective, and further targeted energy assistance may be needed based on a diverse selection of metrics.

Although they are the states with the highest $N_h$, California ($N_h = 59$) and Colorado ($N_h = 63$) are not immune to these problems and likely represent a greater spread and diversity of energy affordability impacts. Likely, this diversity captures the benefits accrued by early adopters and the challenges of having high populations of those struggling with energy poverty and high housing costs. In many of these places, residents have self-sorted into geographic areas based on the overall costs of living. Also, advances in clean energy legislation are a common thread among the top-performing states on an $N_h$ basis, signaling the value of strong decarbonization targets and accompanying policies to ensure electricity affordability for low-income households.

Visualizing the proportion of end-use energy sourced from the combustion of fossil fuels in Fig. 5 shows that such reliance does not necessarily lead to a more affordable system for energy consumers. Households in states with a high proportion of fossil fuel are no less likely to have high $N_h$s than those in other states that rely on clean energy, defying the conventional wisdom that fossil-fuel consumption is a chosen tradeoff between environmental health and affordability for citizens.

**Discussion.** A sensible prior hypothesis is that everyone experiences the same efficiency from the energy system as measured by return on energy investment. Differences in absolute outcomes may be related to the quantity of energy investment, but the marginal unit of energy consumed by one household should lead to as much benefit for that household as any other. However, here we see that $N_h$s are different among different groups of households in the US. This difference is often correlated to factors out of the households' control and even those related to persistent social inequalities, such as race and education. These striking disparities suggest the existence of deeply structural barriers to prosperity in the US. Energy is central to equity and economic prosperity, but the energy system appears to be regressive in that costs accrue disproportionately to those of lower-income levels.

Furthermore, we demonstrate that owning a home and consuming solar power are associated with increased income multipliers for energy expenditures. This advantage leads to gains that are not being realized by many communities. When households adopt solar power, their $N_h$ increases as a result of decreasing their energy expenditures, which creates a disparity

between those with access to renewable energy and those reliant on fossil-fuel-based energy sources. This helps explain why there has been a disparity in how the benefits of the energy transition are accruing among socioeconomic groups[34]. There is the potential for electrification and the transition to clean fuels to exacerbate this division if appropriate policies are not implemented[35].

Indeed, there are clear, mutually synergistic, positive reinforcement mechanisms to alleviate health and environmental disparities in air pollution exposure by reducing household energy burdens and improving economic mobility across low-income households. Combustion of biofuels and hydrocarbons is a significant source of air pollution and exacerbates other household costs like healthcare and maintenance, yet we find that utilization of these sources is not associated with increased $N_h$s at a state or household scale. Not only are households living in more poverty and closer proximity to highly polluted areas at greater risk of adverse health impacts; they must also consume more energy to overcome the particulate emissions, which, themselves, reduce the efficiency of clean sources such as solar panels[36].

The inherent benefits of solar electricity must be accessible to all populations in the US to promote sustainability, but communities of color are not receiving a similar benefit to white and wealthier households from their energy expenditures. Net energy metrics exhibit this income multiplier effect and the resulting divide. Designing solar policies to benefit those facing low $N_h$s may substantially improve net energy income ratios and raise households out of energy poverty in the US.

Pachauri et al. distinguish between affordability and cost of supply, implying that more focus should be applied toward how energy burdens vary among customers of different energy suppliers than on how per-unit costs of energy vary[9]. For instance, Roanoke Electric Membership Corporation is the electric utility with more than 10,000 customers whose members have the lowest average $N_h$ (13), according to LEAD. At the same time, the per-unit cost of electricity in this service territory is only more expensive than 76% of utilities at $0.14/kWh. Is an exceptionally high burden acceptable because the per-unit costs are not exceptional?

This relevance especially holds for electricity because it is a commodity delivered via a stationary, centralized grid system, and households retain little control over their own energy choices. In vertically integrated energy markets, the monopoly utility is the only option available to all consumers. In organized energy markets, the public utility is designated as the last resort provider for those unable or unwilling to participate in the competitive procurement of energy. Even in organized markets, local utilities retain monopolistic control of the transmission and distribution systems.

Consumers are price takers with relatively inelastic demand. Changes in the unit price of energy or slight differences in consumption patterns matter more to those with low incomes than those with higher incomes. Furthermore, the current lack of storage infrastructure on the grid and behind each meter means that households are bound to electricity providers at the time of use. The "forward-looking" or "reactive" tendencies of these public electric utilities have implications for the energy transition in their jurisdictions and beyond[37]. A more in-depth examination of the energy system's underlying regulatory structures and robust assessments of energy burden could provide a path forward and track how the benefits of the energy transition are being accrued.

In some markets, specialized rates or programs are available for Low and Moderate Income (LMI) consumers, who may have higher energy burdens. We find it notable that more than half of all funding to address high energy burdens in the US is from utility ratepayer-funded bill and energy efficiency assistance[14]. At a national scale, the Low Income Home Energy Assistance

Program (LIHEAP) and the Weatherization Assistance Program (WAP) in the US seek to address aspects of energy poverty through bill payment assistance and energy efficiency measures, but the efficacy of these programs has been mixed in addressing distributional equity in energy burdens and receiving benefits from energy efficiency programs[11]. In part, this may be because these programs rely on income-based poverty lines such as the FPL to determine eligibility for benefits.

The US benchmarks its FPL to the food requirements of the average household[38] and uses this threshold as is an eligibility criterion for more than 40 federal programs across ten agencies (in addition to state, charitable, and private enterprises that also do so)[39]. Practitioners have posited that the standard policy of "using the 'economy food plan' to determine who can afford to hire an attorney" may be depriving citizens of their basic rights[40]. In some cases, food assistance programs are more inclusive than energy assistance programs: we show that more than 5.2 million households above the FPL experience energy poverty.

Research using detailed qualitative sociological and public health interview data links energy burden with housing and energy policy[41]. Other articles have linked technology adoption barriers to low-income communities of color[42–44] and identified that households that are in the same peer social network often adopt technologies such as solar[45]. While some studies already acknowledge the benefits of improving energy efficiency and equity outcomes through surveys and interviews[46], this study adds to the literature by building a comprehensive quantitative framework and empirical result based on a large dataset. This toolkit adds more quantitative backing to this body of qualitative work and can be used in future analysis with technology adoption data to identify strategic opportunities to improve energy efficiency and access to clean technology in a more equitable manner.

Previous work provided theoretical frameworks for understanding energy poverty beyond a simple income-based measure[7], and limited examples of applications to empirical data that rely on building-specific efficiency characteristics[12] or subjective self-assessments[13]. Applying $N_h$ to a nationwide dataset for the US builds on this work by providing a tangible tool that includes 5.3 million households with energy expenditures greater than their incomes or with incomes above the FPL that are left out by other measures, and is a way to visualize the disparities between groups that may allow for further investigation around different household characteristics. $N_h$ highlights inequality better than a simple linear metric while allowing the assessment of a wide variety of households, such as those with no income or for which efficiency data is not available. A significant insight from NEA that should be incorporated into future work is that embodied energy takes many forms across the household budget (food, goods, services, transportation, housing, etc.), and these can all be compared using the same units of measure (e.g., joules) to scale the $N_h$ framework across multiple expenditure categories.

The Coronavirus Disease 2019 (COVID-19) pandemic and associated recovery measures may be a critical opportunity to provide relief payments related to energy expenditures and to invest in more efficient residential[47] and commercial energy infrastructure that enables newer and cleaner systems[48]. We show that US households are already spending excessive amounts on energy, notwithstanding more families staying at home for longer periods of time during pandemic lockdowns. The ongoing crisis offers a chance to address inequity with a focus on residential energy burdens[49]. The EU is in the process of defining how communities can participate in the energy transition[50] and how burdens can be alleviated through this process using proactive policy tools and business models such as One Stop Shops (OSS)

for energy efficiency and renewable energy upgrades[51]. Adopting energy criteria for energy and other programs like these could expand access for those underserved populations in need of assistance independent of their needs in other consumption categories.

Creating a federal energy poverty line would be a critical step in identifying families that face large disparities in access to affordable electricity and energy in the US and improve programs' abilities to address energy burdens. A toolkit based on this analysis enables neighborhood level outreach where burdens are highest and identifies opportunities where households could benefit from emerging technologies.

## Methods

**Data**. To estimate the Net Energy Return ($N_h$) of American households, we primarily utilize the Low Income Energy Affordability Data (LEAD)[23] and Rooftop Energy Potential of Low-Income Communities in America (REPLICA)[52] datasets, which the DOE assembled to help "stakeholders make data-driven decisions on energy goal setting and program planning by providing them information on low-income household populations and associated energy use characteristics"[23]. These datasets encompass estimates of household energy expenditures (S), income (G), and demographic characteristics for most households at the census-tract scale in all states and most territories of the US.

LEAD: The LEAD portrays the average income, electricity expenditures, gas expenditures, and other fuel expenditures for cohorts of households segmented by location (census tract, county, state) and household characteristics (whether the unit is rented or owned, the building's year of first construction, the number of units in the building, whether the units are attached, and the unit's primary heating fuel type). The dataset is assembled by applying an iterative proportional fitting (IPF) algorithm to cross-tabulations of household responses from the 2016 5-year American Community Survey (conducted by the US Census Bureau), which provides the samples for each cohort as Public Use Microdata Samples. IPF is a widely used spatial microsimulation method to allocate individuals (i.e., households) to zones (i.e., census tracts and utility service territories) while calibrating each zone's characteristics to known quantities. Using IPF, the microdata samples are scaled to match aggregate annual values from utility sales and revenues reported in Energy Information Administration forms 861 and 176.

REPLICA: The Renewable Energy Potential of Low-Income Communities in America (REPLICA) dataset includes the racial and education level composition of each census tract used in the ultimate analysis[52]. In addition to providing a simpler designation of cohorts for each census tract, REPLICA also includes estimates of the technical potential of rooftop solar and additional techno-economic variables (e.g., demographics and electricity rates) that will be useful for future research.

eGRID: We use the EPA's eGRID[53] to calculate the proportion of household energy expenditures that support fossil-fuel combustion through the purchase of electricity. 100% of natural gas purchases are considered to support fossil-fuel combustion. Electricity purchases are divided into their respective sources according to the state proportions indicated by the "STCLPR" (coal), "STOLPR" (oil), "STGSPR" (gas), and "STOFPR" (other fossil) fields in the 2018 eGRID dataset. Other expenditures are divided according to the primary heating fuels other than electricity or natural gas used by American households according to the Census, which are approximately 76% fossil-fuel combustion.

**Treatment**. The LEAD data represents the unit's ownership status (OWNER vs. RENTER) and income bracket as a fraction of AMI (0–30%, 30–60%, 60–80%, 80–100%, or 100%+) or FPL (0–100%, 100–150%, etc.). These categorical variables are saved as factors. Then we create min_units from BLD INDEX, a variable which represents a non-uniformly distributed set of buckets for the range of the number of units in the building and whether single-unit households are attached or detached from neighboring households (1 ATTACHED, 1 DETACHED, 2 UNIT, 3-4 UNIT, 5-9 UNIT, 10-19 UNIT, 20-49 UNIT, 50+ UNIT, MOBILE_TRAILER, BOAT_RV_VAN, OTHER UNIT). We extract the minimum number of units from the range and whether the building is detached. Those households labeled OTHER UNIT, MOBILE_TRAILER, or BOAT_RV_VAN are given values of Not Applicable (NA) for this characteristic. Finally, we calculate S and G of which each metric is composed:

S = annual expenditures on electricity (ELEP CAL) + natural gas (GASP CAL) + and other fuels (FULP)

G = the cohort's average annual income (HINCP)

The metric formulas outlined in Applying NER to Energy Equity are then used to calculate each cohort's energy poverty metrics. Since we are examining homes' relationships with the energy system, we ignore any homes that do not use energy as denoted by rows where S == 0. The estimation procedure used by the DOE results in an estimated number of occupied housing units for each cohort (UNITS, renamed as households). It displays the number of American Community Survey responses that contribute to the estimate of energy expenditures (COUNT,

renamed as acs_responses). We then remove any categories with fewer than 1 unit represented since this is not physically possible.

We then combine this dataset with the REPLICA dataset. To do so, we must aggregate the income levels of the LEAD dataset to the simpler schema used by REPLICA for summarizing households' income relative to the area's median income (AMI):

- 0–30% AMI: Very-Low-Income
- 30–80% AMI: Low-to-Moderate-Income
- ≥ 80% AMI: Middle-to-High-Income

Also, we create an indicator of whether a particular cohort is in income poverty as defined by the relevant standards for its characteristics. For the AMI version of LEAD, this is defined as being "Very-Low-Income" or ≤ 30% of AMI. For the FPL version of LEAD, we translate directly from the designation of the income bracket as follows:

- 0-100% FPL: In Poverty
- ≥ 100% FPL: Not In Poverty

The REPLICA dataset also simplifies any households with only one unit per building into "Single Family" homes and any households with more than one unit per building as "Multi-family." Non-stationary and non-traditional homes are not included in the REPLICA analysis. We match these simplifications in the LEAD dataset by aggregating by the number of units:

- 1 Unit: Single-Family
- > 1 Unit: Multi-Family
- Other Unit: NA (excluded from analysis)

After simplifying these characteristics in the LEAD AMI data, we merge the AMI dataset with the REPLICA dataset along the census tract, simplified income bracket, simplified number of units, and housing tenure variables to achieve the primary dataset used in the analysis. Merging with the REPLICA dataset provides additional demographic and geospatial data not available in the LEAD dataset only[52]. Characteristics such as the utility type and locale description are sourced from REPLICA and unavailable in LEAD, and we source the cost of electricity, racial composition, and education levels of census tracts from REPLICA for this analysis. Similarly, the REPLICA dataset contains only electrical expenditure estimates without natural gas or other fuel costs to households and could not be used to perform this analysis alone.

The FPL version of LEAD is not merged with all of the REPLICA data because of incompatibility between the FPL and AMI bracket definitions. However, demographic data associated with each census tract as a whole can be merged with the FPL dataset. Both AMI and FPL versions of LEAD are combined with demographic data from the REPLICA dataset and geospatial shapefiles from the Census to produce the final datasets used in the analysis. Since the FPL version of LEAD is not aggregated to the simpler categories found in REPLICA, more granular variables such as Primary Heating Fuel remain available for assessment across the entire population.

**Considerations**. While a helpful place to start, $E_b$ has certain drawbacks. Significantly, a simple proportion does not account for the fact that money spent on energy cannot be spent elsewhere and is therefore not useful income to the household for the purposes of measuring prosperity. Including gross income in $E_b$ has the effect of depressing the average $E_b$, by definition.

Furthermore, because energy expenditures are a small proportion of even the most impoverished households' total income, $E_b$ is almost always a very small percentage (<10%). This leads to issues with interpretability in public discourse and policy settings and may even affect program outcomes that are based on $E_b$ if small numbers are rounded to even the nearest hundredth of a percent. As the UK experienced when using $E_b$, if the household is above an energy poverty line defined by such a metric for public policy purposes, the family may not receive critical support[11]. As the UK experienced when using

Because such a small proportion of expenditures can impact different income groups so differently, ratios of this type can be useful when delineating across income quantiles or other categories - particularly for vulnerable populations where energy poverty poses a significant difficulty or affordability threshold not captured by measures of absolute poverty. However, $E_b$ is often portrayed at a population scale (e.g., the average energy burden of the population is X%), which can be skewed by outliers within the population. Household metrics and surveys are important for further understanding of and policy development around issues of energy poverty.

Finally, presenting the relationship between household income and energy expenditures as a proportion with income in the denominator suggests that improvements can only be made by decreasing energy expenditures or increasing incomes. However, in reality, there is a positive relationship between energy expenditures and incomes because energy is an input to wealth-creating processes. Households consume energy to unlock the utility that energy services provide to them as participants in society, whether to cook food or connect to the internet. This relationship is generally accepted when understanding individual household behavior[54] and macroeconomic effects of energy consumption[55]. Therefore, a metric describing the efficiency of household wealth creation may be more intuitive with income in the numerator.

The iterative proportional fitting method has limitations as an estimation procedure that constrain the strength of conclusions that can be drawn from the simulated LEAD and REPLICA datasets. The relationship between constraint variables such as total energy spending by utility service territory and number of households per census tract will tend toward the average of the initializing dataset and depress variations among otherwise similar regions. This may explain the large quantities of households that are estimated to have very low incomes. Validating these estimated data would require randomized surveys of households along the dimensions of interest.

The meaning of the "primary heating fuel" category comes from the answer to the question "Which fuel is used most for heating this house, apartment, or mobile home?" on the American Community Survey. This question's power to predict energy expenditures or fuel sources is unknown, and further analysis is required to understand the implications of this survey question for drawing broader conclusions about household energy use. However, the US Census Bureau has been asking this question since 1940. It states that these data are collected to help communities "provide assistance with utilities," "estimate future energy demand," and "measure environmental impacts"[56].

Though the REPLICA dataset relies on a different vintage of the LEAD dataset (assembled in 2017[52]) than this analysis (assembled in 2019[23]), inferring differences among annual estimates is not meaningful due to the standard error of the data[23]. A rigorous treatment of these metrics over time is an area for future research.

The results inferred from eGRID are only as good as the eGRID methodology itself. We choose to outline the proportion of combusted fuels rather than emissions data due to the limitations outlined by proponents of marginal emissions attribution in electric power system models[57,58] and limitations in estimating regional electricity transfers[59].

**Reporting summary**. Further information on research design is available in the Nature Research Reporting Summary linked to this article.

## Data availability

The data generated in this study have been deposited in the Zenodo database under https://doi.org/10.5281/zenodo.5676225[60].

All data necessary for the composition of the source datasets used in this analysis are freely available from US government sources as open data. All functions to automatically retrieve and assemble these data and compiled versions of these data are made available to the user as part of the software available at https://doi.org/10.5281/zenodo.5676225[60].

## Code availability

The code and data to fully reproduce this paper are available on GitHub at https://github.com/ericscheier/net_energy_equity under the GNU Affero General Public License v3.0. The most recent release of the software can be found at https://doi.org/10.5281/zenodo.5676225[60].

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

## Acknowledgements
The authors wish to thank Nikhil Kaza and Andy Yates for insightful conversations, review, comments, and feedback on this study.

## Author contributions
E.S. and N.K. conceived of and designed the project, wrote, and edited the paper. E.S. created the software used for data acquisition and analysis and created the figures. N.K. contributed the introduction of the topic, interpretation of the results, and supervised the study.

## Competing interests
The authors declare no competing interests.
