## [Peer Review File · Nature Communications]

The paper “*Net energy metrics reveal striking disparities across United States household energy burdens*” investigates a topic of high-policy relevance, namely the measurement of energy poverty in the United States (US) and how it is distributed across different social groups. In particular, the authors investigate the relationship between household income for all census tracts in the US and energy spending.

The authors apply the analysis of net-energy return (as the dollars earned for every dollar spent on energy) and the energy burden (as the proportion of income spent on energy) to one of the common definitions of energy poverty (intended as an energy burden that is an expenditure greater than 10% of the household income), and derive the energy poverty line accordingly.

The authors show that accounting for different household income levels within census tracts provide a picture of the energy inequity that exists across different social groups who live in specific geographical areas (such as rural areas or those close to the US-Mexico borders).

Controlling for the most prominent demographics in the census tract of each cohort, such as the level of education, race, and housing tenure, the authors show that further disparities related to energy burdens exist. As an example, homeowners experience a higher energy return than tenants.

The proposed framework not only enables to better track the magnitude of energy poverty in the US, but also unveils the justice concerns that policy makers should take into account when looking at the differential impact of energy poverty across different social groups.

1. The study not only adds to the literature on the measurement of energy poverty, but also on energy justice, as it provides a framework enabling to unveil how lower-income and ethnic groups are impacted by energy burdens, highlighting the need to take into account justice concerns in energy policy-making.

To better highlight these contributions (and also better clarify what the proposed framework does not address, such as the “hidden energy poor”), the authors might want to provide more background on the measurement approaches (and definitions) of energy poverty (as an example, see Faiella and Lavecchia 2021), on the multidimensional approaches (see Sokołowski et al. 2020), as well as the distributional justice aspects of energy justice (see Sovacool and Dworkin, 2015).

Faiella, Ivan, and Luciano Lavecchia. "Energy poverty. How can you fight it, if you can't measure it?." *Energy and Buildings* 233 (2021): 110692.

Sokołowski, J., Lewandowski, P., Kietczewska, A., & Bouzarovski, S. (2020). A multidimensional index to measure energy poverty: the Polish case. *Energy Sources, Part B: Economics, Planning, and Policy*, 15(2), 92-112.

Sovacool, Benjamin K., and Michael H. Dworkin. "Energy justice: Conceptual insights and practical applications." *Applied Energy* 142 (2015): 435-444.

2. The authors might also want to add references to studies that might further strengthen their claims.
 - lines 166-170 the authors refer to lower-income households' low consumption. The authors might want to refer to the prebound effect introduced by Sunikka-Blank and Galvin 2012.
 - The authors talk about assistance or energy efficiency programs, but fail to mention other measures or initiatives that might contribute to tackle energy poverty. This is especially the case of energy communities (see Caramizaru and Uhlen 2020) and of one-stop-shops for energy-renovations (see Bertoldi et al. 2021), which also have the potential to promote vulnerable groups' agency.
 - lines 67-68 The authors might also want to refer to the capabilities approach applied to energy poverty (Day et al. 2016).
 - lines 289-291, the authors mention that several barriers hold back some communities from taking full advantage of solar electricity. The authors might also want to mention that lower-income households or minority ethnic groups are also more vulnerable to the impact of air pollution, which, among others, reduces the actual electricity generated by solar panels (see He et al. 2020).

Bertoldi, Paolo, et al. "The role of one-stop shops in energy renovation-A comparative analysis of OSSs cases in Europe." *Energy and Buildings* (2021): 111273.

Caramizaru, Aura, and Andreas Uihlein. *Energy communities: an overview of energy and social innovation*. Publications Office of the European Union, 2020.

Day, Rosie, Gordon Walker, and Neil Simcock. "Conceptualising energy use and energy poverty using a capabilities framework." *Energy Policy* 93 (2016): 255-264.

He, P., Liang, J., Qiu, Y. L., Li, Q., & Xing, B. (2020). Increase in domestic electricity consumption from particulate air pollution. *Nature Energy*, 5(12), 985-995.

Sunikka-Blank, Minna, and Ray Galvin. "Introducing the prebound effect: the gap between performance and actual energy consumption." *Building Research & Information* 40.3 (2012): 260-273.

Overall, the conclusions and claims are supported by a well-conducted analysis. Limitations are clearly stated in the methodology, implicitly providing avenues for future research.

3. Some minor points:

- The authors should adopt the acronyms consistently in the text (once introduced, they should use them).
- When discussing the analysis, the authors should refer in text to the related figure/table.
- Some typos often occur throughout the text.

Reviewer #2:

Remarks to the Author:

This article investigate the energy burden across US households. Here the authors present a national analysis aimed at highlighting who is experiencing energy poverty.

The abstract needs clarification. In the line 11 – 16 on page 1 seem to be off. It says just 0.29% of homes above the federal poverty line also face energy poverty. This number seems underwhelming compared to the 78% of those below the federal poverty line. When the abstract concludes with needing to expand program to capture the 0.29% of people it is unclear what type of impact this would have. While it is valid to say that the federal poverty line cannot capture all of energy poverty I think the authors should mention some of the other things like risk of outages see Graff et al 2021 <https://doi.org/10.1016/j.erss.2021.102144> and people's inability to use energy see Cong et al 2021 <https://www.researchsquare.com/article/rs-712945/v1>

Line 23 – 29 of page 1. I think this would benefit from more specifics. What s causing energy to become more unaffordable? When you say most households don't have access to energy tech how many is that? What %?

In the first paragraph of the introduction you discuss infrastructure barriers to adoption, then in the next you discuss costs. I think these paragraphs could be tied together better by saying that there are multiple limitations to low-income communities adopting cutting-edge energy technologies. First infrastructure barriers.....In addition there are cost barriers.

For the equations they should have the variables referenced in the text, and the only have the equation stand out from the text. Also if you use S for multiple things it should have subscripts for each different variable meaning. For example you have S and S_income. Also for equations it will make it easier to understand if you have one letter per variable. For example when reading it some people may thing NER is three variable multiplied together. You could just name it N subscript h, and have the subscript I represent income.

The NER is defined twice (as equation 2 and 4). This is redundant, and should be combined in the text. I think you should explain the equations in relevant text.

There should be a discussion of limitations of energy burden. For example it cannot capture who is not using energy (i.e. those who have received power cuts)

For Figure 1, I am not seeing the main takeaway. What do we learn that is new to the literature? How do you define net income? I was surprised to see that most households had net income less than 60,000. For figure 1 it seems like it would be valuable to zoom in on the 0 – 100% It feels odd to see the x axis extend all the way to 400%. I understand that this is because those households may not have an annual income, but it still feels misleading. Since you use the circle sizes I wonder if it would be useful to have a circle that says > 4 million.

Please move the AMI acronym definition to when the AMI is first mentioned.

What is equation 7 for? All equations should be referenced in the text and explained. Without this is leaves the reader wondering what the purpose of the information is.

In general it is good practice to discuss the figure in the text before it is presented. When talking about the households in figure 4 do you mean that 10% of census tracts experience energy poverty? Also, the discussion about most households who are experiencing energy poverty also experiencing energy poverty seems misleading. Because energy poverty is a function of income it follows that the two would be correlated. Since they are not independent of each other it makes sense that if you are experiencing one then you are likely to experience the other.

In the introduction there was a lot of discussion about technology adoption, so I was surprised that this was not discussed in detail (with a figure) in the main text. Based on the introduction framing I experienced there to be some analysis on which communities are experiencing technology adoption barriers. I see in your methods you use the Rooftop Energy Potential of Low-Income Communities in America. I was not sure how this fed into your analysis. In the methods it would have been helpful to mention the equations this feeds into, and more detail about what it is used for in your main text analysis. Without this information it will be hard for others to replicate this work, and build on your analysis.

In general I think this paper covers an important topic (energy poverty, and who is experiencing that poverty), but I think the paper is lacking in the insights. One thing I would have liked to see is more detailed description of what value this paper adds to the multitude of studies that examine energy poverty in the US. Additionally, it seems that the paper was discussing households, but the data appears to be at the census tract level. Lastly, I think the methods needed more clarity. Some of the variables were defined twice, and some equations lacked explanations in the text which made the article hard to follow. I hope my comments above are helpful to the authors as they continue down an important research path.

Reviewer #3:

Remarks to the Author:

This study has many promising and thought-provoking components, but it suffers from a bit of an "identity crisis." As written, the manuscript jumps around and lacks a cohesive narrative, and I think attempts to do too much in a short amount of space. As a consequence, the potential contributions of the study are masked and the data and analysis presented is under-developed.

As an example of the confusing presentation, the introduction begins with a discussion of energy affordability, then moves into a micro-level discussion of limitations of the FPL before jumping back to macro-level discussion of the implications of energy affordability for social cohesion, climate change, and sustainable development, before providing a one paragraph summary of the empirical analysis presented in the paper followed by a general statement about facilitating a "sustainable and prosperous society built on equity and justice." At this point as a reader, it is very difficult to know what the paper is about. Rewriting can certainly fix this, but the disjointed presentation points to a more general problem with the current manuscript which is that it is difficult to follow.

The main potential contribution of the study is the application of "net energy return" as an indicator of household energy poverty, but it is not fully-developed in the current manuscript. This is an intriguing idea, and the authors offer several reasons for why this measure has analytical advantages compared to the more traditional measure of energy burden. The paper would benefit from a more robust discussion here. In fact, the text in the caption to Figure 1 is quite enlightening, and I think can be moved to the main text to amplify the arguments. More generally, the authors should give more attention to this part of the manuscript. To make room given the limited word count, I would suggest the authors substantially cut back the discussion section, much of which is speculative and not supported by the analysis provided (more on this below).

A second potential contribution of the paper is the analysis showing the spatial distribution of NER and its composition by socio-demographic characteristics. Here too, the manuscript would benefit from more detailed discussion of what Table 1 and Figures 2, 3, and 4 demonstrate. These graphics each present a large amount of data, but it is not obvious what a reader is supposed to take away from them. In addition, new metrics and data sources are introduced but not explained (e.g., Area Median Income, LEAD); the details are in the Methods section but the authors need to provide enough explanation in the main text so the reader can follow along.

In addition to a more complete discussion of the main contributions of their study, the authors in my view need to be much more careful in their interpretation. The analysis contains a number of assertions that neither directly follow from the data presented nor are supported with appropriate citations. For instance, in the last full paragraph on p.4, the authors make several assertions about fossil fuel costs, environmental and health externalities, etc. without evidence. The next partial paragraph similarly makes a statement about areas where inexpensive energy is available, and about housing quality and energy consumption, none of which come directly out of the data presented. Later in this section, the authors discuss renters v. owners and households with solar power, but again this analysis does not follow directly from the data, or at least, not the way it is presented. Similar problems emerge in the discussion section, where the authors speak of energy efficiency, intermittent grid access, utility disconnections, in-house combustion of biofuels and hydrocarbons, etc. This list goes on to include loans for solar programs, vertically integrated

markets, the Roanoke Electric Membership Cooperation, and LIHEAP/WAP, etc.

In sum, too much of the discussion is disconnected from the data and analysis presented, which is quite frustrating from a reader's standpoint. More importantly, I think the authors do a disservice to themselves, because the scattered and speculative discussion takes away from an opportunity to better highlight the important contributions of the research. A broader discussion is certainly appropriate, but I might suggest a different approach in which the authors engage more deeply with other studies of household energy poverty, energy burden, and energy insecurity to explain what their measurement approach brings to the literature and to our understanding of these conditions.

Some more specific comments:

p.1, the problem characterized as a principal-agent dilemma is more commonly referred to as the split-incentive problem.

p.1, the assertion that "bureaucracy and institutional inertia" are holding back a rapid transition is vague and not substantiated.

p.2 the authors state that the study provides "a biophysical framework to evaluate the disparities among household net energy outcomes", but this is not explained, and I frankly do not understand what the authors mean.

p.3 The authors assert that energy poverty is commonly defined as a household spending 10% or more of its income on energy. The citation is to Bednar and Reames (2020), but they are referring to fuel poverty measurement in the UK. In the US, I think 6% is more common, as in this report from ACEEE (<https://www.aceee.org/sites/default/files/energy-affordability.pdf>).

NCOMMS-21-25017A

Net energy metrics reveal striking disparities across United States household energy burdens

Line numbers and citations refer to the cleaned version of the review (without tracked changes) unless otherwise noted.

We thank all reviewers for their thorough and detailed remarks to improve this manuscript.

REVIEWER COMMENTS

Reviewer #1 (Remarks to the Author):

Reviewer: The paper “Net energy metrics reveal striking disparities across United States household energy burdens” investigates a topic of high-policy relevance, namely the measurement of energy poverty in the United States (US) and how it is distributed across different social groups. In particular, the authors investigate the relationship between household income for all census tracts in the US and energy spending.

The authors apply the analysis of net-energy return (as the dollars earned for every dollar spent on energy) and the energy burden (as the proportion of income spent on energy) to one of the common definitions of energy poverty (intended as an energy burden that is an expenditure greater than 10% of the household income), and derive the energy poverty line accordingly. The authors show that accounting for different household income levels within census tracts provides a picture of the energy inequity that exists across different social groups who live in specific geographical areas (such as rural areas or those close to the US-Mexico borders). Controlling for the most prominent demographics in the census tract of each cohort, such as the level of education, race, and housing tenure, the authors show that further disparities related to energy burdens exist. As an example, homeowners experience a higher energy return than tenants. The proposed framework not only enables to better track the magnitude of energy poverty in the US but also unveils the justice concerns that policymakers should take into account when looking at the differential impact of energy poverty across different social groups.

Response: We thank the reviewer for their valuable comments and suggestions to improve this analysis and the communication of our results. To date, there are no net-energy return studies we are aware of that focus on energy burdens or energy poverty, yet this metric may be valuable for a greater understanding of energy burden characteristics and highlight energy inequity challenges that the US faces.

Reviewer: The study not only adds to the literature on the measurement of energy poverty, but also on energy justice, as it provides a framework enabling to unveil how lower-income and ethnic groups are impacted by energy burdens, highlighting the need to take into account justice concerns in energy policy-making.

To better highlight these contributions (and also better clarify what the proposed framework does not address, such as the “hidden energy poor”), the authors might want to provide more background on the measurement approaches (and definitions) of energy poverty (as an example, see Faiella and Lavecchia 2021), on the multidimensional approaches (see Sokołowski et al. 2020), as well as the distributional justice aspects of energy justice (see Sovacool and Dworkin, 2015).

Faiella, Ivan, and Luciano Lavecchia. "Energy poverty. How can you fight it, if you can't measure it?." *Energy and Buildings* 233 (2021): 110692.

Sokołowski, J., Lewandowski, P., Kielczewska, A., & Bouzarovski, S. (2020). A multidimensional index to measure energy poverty: the Polish case. *Energy Sources, Part B: Economics, Planning, and Policy*, 15(2), 92-112.

Sovacool, Benjamin K., and Michael H. Dworkin. "Energy justice: Conceptual insights and practical applications." *Applied Energy* 142 (2015): 435-444.

Response:

We expanded the literature review and included these key studies in the background and discussion to provide foreground context into the existing work on energy justice and tie that to our contribution using net energy return. For example, please see highlighted sections in the tracked changes introduction.

[Lines 34-53] “Many frameworks are currently being explored to understand this issue. In a thematic exploration of energy equity, Brown et al. identify energy access, energy poverty, energy insecurity, and energy burden as key concepts for understanding the issue⁸, but measurement of these concepts has been limited to date. Pachauri & Rao establish measures for the sustainable development context that incorporate periods when energy is available, the quality of voltage supplied, the reliability in terms of the number of disruptions, the capacity in terms of power available, the consumption levels allowed per day, and affordability of the standard consumption package as a percentage of household income⁹. Energy metrics have been assessed quantitatively across several countries. Though some aspects, such as formal

disconnections from energy service¹⁰, are translatable to the US context, these methods require normalizing many variables amongst different types of data and are overly broad for applications in areas where electricity access is relatively high and reliable.

Of such areas, the United Kingdom (UK) has a richer history of incorporating energy poverty formally into its government programs: since 2000, the UK has used some form of an energy burden metric to assess whether households are facing energy poverty and determine the level of support that they require as a result¹¹. This metric has evolved from a simple ratio of household income and energy expenditures to one that incorporates building efficiency ratings and average incomes in the community. While the European Union (EU) currently lacks a unified metric for energy poverty, similar metrics have been developed in member countries, such as a metric which compares incomes and expenditures to local averages and absolute heating needs to determine energy poverty levels in Italy¹² or a multidimensional index of building quality and ability to pay bills in Poland¹³.”

Reviewer: The authors might also want to add references to studies that might further strengthen their claims. [In] lines 166-170 the authors refer to lower-income households’ low consumption. The authors might want to refer to the prebound effect introduced by Sunikka-Blank and Galvin 2012.

Sunikka-Blank, Minna, and Ray Galvin. "Introducing the prebound effect: the gap between performance and actual energy consumption." *Building Research & Information* 40.3 (2012): 260-273.

Response:

We thank the reviewer for suggesting these relevant and valuable studies that strengthen our claims.

We added references that address the prebound effect in the context of lower-income households’ low consumption, as recommended, and further relevant references.

[Lines 224-227] “Why are certain households not receiving the same benefits of their fuel source across the distribution of incomes? This could be due to lower-income households’ low consumption, meaning that the potential savings from implementing energy efficiency measures are lower than for high-income households according to the prebound effect identified by Sunikka-Blank and Galvin³² and Cong et al.³³.”

Reviewer: - The authors talk about assistance or energy efficiency programs, but fail to mention other measures or initiatives that might contribute to tackle energy poverty. This is especially the

case of energy communities (see Caramizaru and Uhlen 2020) and of one-stop-shops for energy-renovations (see Bertoldi et al. 2021), which also have the potential to promote vulnerable groups' agency.

Caramizaru, Aura, and Andreas Uihlein. Energy communities: an overview of energy and social innovation. Publications Office of the European Union, 2020.

Bertoldi, Paolo, et al. "The role of one-stop shops in energy renovation-A comparative analysis of OSSs cases in Europe." *Energy and Buildings* (2021): 111273.

Response:

In this revision, we include other programs around the world, particularly in the EU, which have advanced more social programming than the US related to energy poverty. The European Union's initiative to address energy poverty and the One Stop Shop model are examples that could also be applied to address the issue of energy inequity in the US.

[Lines 368-373] "The EU is in the process of defining how communities can participate in the energy transition⁵³, and how burdens can be alleviated through this process using proactive policy tools and business models such as One Stop Shops (OSS) for energy efficiency and renewable energy upgrades⁵⁴. Adopting energy criteria for energy and other programs like these could expand access for those underserved populations in need of assistance independent of their needs in other consumption categories."

Reviewer: lines 67-68 The authors might also want to refer to the capabilities approach applied to energy poverty (Day et al. 2016).

Day, Rosie, Gordon Walker, and Neil Simcock. "Conceptualising energy use and energy poverty using a capabilities framework." *Energy Policy* 93 (2016): 255-264.

Response:

Thank you. The capabilities approach is an important contribution to the energy poverty framework. We cite this particular study to emphasize that energy poverty is more than just a lack of money.

[Lines 26-27] "Furthermore, energy inequity is not just a lack of money to meet basic energy needs - it is a lack of access to the capabilities⁶ that enable a sustainable and prosperous society built on just principles⁷."

Reviewer: - lines 289-291, the authors mention that several barriers hold back some communities from taking full advantage of solar electricity. The authors might also want to mention that lower-income households or minority ethnic groups are also more vulnerable to the impact of air pollution, which, among others, reduces the actual electricity generated by solar panels (see He et al. 2020).

He, P., Liang, J., Qiu, Y. L., Li, Q., & Xing, B. (2020). Increase in domestic electricity consumption from particulate air pollution. *Nature Energy*, 5(12), 985-995.

Response: This is a great point and we thank the reviewer for highlighting this type of work on air pollution, which is also highly relevant to energy burden analysis. We have included this study as an example in the discussion of particulate emissions' impacts on vulnerable communities.

[Lines 298-301] “Not only are households living in more poverty and closer proximity to highly polluted areas at greater risk of adverse health impacts; they must also consume more energy to overcome the particulate emissions which, themselves, reduce the efficiency of clean sources such as solar panels³⁹.”

Reviewer: Overall, the conclusions and claims are supported by a well-conducted analysis. Limitations are clearly stated in the methodology, implicitly providing avenues for future research.

Response: Thank you very much, we appreciate the reviewer's remarks and plan to continue in this research path.

Reviewer: Some minor points:

The authors should adopt the acronyms consistently in the text (once introduced, they should use them).

Response:

In this revision we have strived to ensure that any acronyms introduced are used subsequently throughout the text, and that these are defined on their first usage.

Reviewer: When discussing the analysis, the authors should refer in text to the related figure/table.

Response:

This revision elaborates on the discussion of key figures and information with more frequent references to the relevant figures to better guide the reader and interpret this data.

Reviewer: Some typos often occur throughout the text.

Response:

The manuscript has been rechecked for typos, spelling issues, and grammar. These edits are marked in the red tracked changes version of the text.

Reviewer #2 (Remarks to the Author):

Reviewer: This article investigates the energy burden across US households. Here the authors present a national analysis aimed at highlighting who is experiencing energy poverty. The abstract needs clarification. In the line 11 – 16 on page 1 seem to be off. It says just 0.29% of homes above the federal poverty line also face energy poverty. This number seems underwhelming compared to the 78% of those below the federal poverty line. When the abstract concludes with needing to expand program to capture the 0.29% of people it is unclear what type of impact this would have.

Response:

The abstract is rephrased to clarify this number - approximately 5.2 million households would be affected by this lack of counting, which is probably a more helpful number and provides better context. We thank the reviewer for pointing this out and helping contextualize this information and impact.

[Lines 14-15] “While 94% of households below the Federal Poverty Line also face energy poverty, more than 5.2 million households above the Federal Poverty Line face this scarcity.”

Reviewer: While it is valid to say that the federal poverty line cannot capture all of energy poverty I think the authors should mention some of the other things like risk of outages see Graff et al 2021 <https://doi.org/10.1016/j.erss.2021.102144> and people's inability to use energy see Cong et al 2021 <https://www.researchsquare.com/article/rs-712945/v1>

Response:

That is a great point and these ideas in particular are very useful. Risk of outages and inability to use energy are more technical energy indicators that would not appear in a traditional income-based economic metric or analysis. These characteristics of energy poverty also warrant consideration to examine energy inequities. In conjunction with Reviewer #1’s suggestions, we have added additional aspects of energy poverty such as the risk of outages, energy capability, and the inability for some people to use energy.

For example, the text now includes these ideas in the following excerpts:

[Lines 41-44] “Energy metrics have been assessed quantitatively across several countries. Though some aspects, such as formal disconnections from energy service¹⁰, are translatable to the US context, these methods require normalizing many variables amongst different types of

data and are overly broad for applications in areas where electricity access is relatively high and reliable.”

[Lines 224-227] “Why are certain households not receiving the same benefits of their fuel source across the distribution of incomes? This could be due to lower-income households’ low consumption, meaning that the potential savings from implementing energy efficiency measures are lower than for high-income households according to the prebound effect identified by Sunikka-Blank and Galvin³² and Cong et al.³³.”

Reviewer: Line 23 – 29 of page 1. I think this would benefit from more specifics. What is causing energy to become more unaffordable? When you say most households don’t have access to energy tech how many is that? What %?

Response:

Incomes have stagnated while housing costs and retail electricity prices are increasing. Household gas and heating costs are also increasing.

[Lines 64-66] “While per-unit residential energy prices have increased below the rate of inflation in the US since the 1980s²², many households still struggle to make utility bill payments and are especially vulnerable to economic shocks²¹.”

With climate change, there could be more demand for heating and cooling, while also lagging investments in energy efficiency for buildings may also explain some of the gap in net energy return observed between households that are rented and owner occupied.

[Lines 238-240] “Renters face systemic disadvantages in the energy transition; they typically pay the home’s energy costs while the landlord controls infrastructure upgrades, commonly understood as the split incentive problem³⁶.”

The data show a gap between renters and owners. More detailed empirical analysis of the drivers of energy costs would also be helpful, yet outside the scope. This study presents the energy burden and identifies spatial, racial, housing tenure, and educational disparities using net energy return.

We removed most text related to new technology adoption at the suggestion of the reviewers, opting instead for a more focused study of the new energy poverty metric we propose. In future work, we hope to connect this research to concepts such as solar adoption rate in the US. However, we did add more specifics to support the assertion regarding adoption:

[Lines 228-232] “Despite technological progress and rapid declines in technology costs and availability for cleaner, renewable electricity generation options such as solar photovoltaics and energy storage, many households cannot take advantage. Most households do not have access to advancements such as low-cost rooftop solar or energy efficiency upgrades that improve air quality, lower greenhouse gas emissions, and save money³⁴: only 18% of households that have adopted rooftop solar have been below the median household income in the US³⁵.”

Reviewer: In the first paragraph of the introduction you discuss infrastructure barriers to adoption, then in the next you discuss costs. I think these paragraphs could be tied together better by saying that there are multiple limitations to low-income communities adopting cutting-edge energy technologies. First infrastructure barriers.....In addition there are cost barriers.

Response:

This language has been removed in favor of a more focused discussion of the energy poverty metrics assessed in this and other related literature. We look forward to incorporating this feedback into future works directed specifically at the adoption of infrastructure to address energy poverty.

Reviewer: For the equations they should have the variables referenced in the text, and the only have the equation stand out from the text. Also if you use S for multiple things it should have subscripts for each different variable meaning. For example you have S and S_income. Also for equations it will make it easier to understand if you have one letter per variable. For example when reading it some people may think NER is three variable multiplied together. You could just name it N subscript h, and have the subscript I represent income.

Response:

The equations and variables are now simplified so that there are only 4 variables (S, I, N_h, and E_b) and 3 equations, which are now described with more detail in the text.

Reviewer: The NER is defined twice (as equation 2 and 4). This is redundant, and should be combined in the text. I think you should explain the equations in relevant text.

Response:

Redundant definitions have been removed and the equations are explained with more detail in the text.

Reviewer: There should be a discussion of limitations of energy burden. For example it cannot capture who is not using energy (i.e. those who have received power cuts)

Response:

We have included a discussion of this point in the introduction and discussion, as well as where it may help explain counterintuitive findings.

[Lines 34-53] “Many frameworks are currently being explored to understand this issue. In a thematic exploration of energy equity, Brown et al. identify energy access, energy poverty, energy insecurity, and energy burden as key concepts for understanding the issue⁸, but measurement of these concepts has been limited to date. Pachauri & Rao establish measures for the sustainable development context that incorporate periods when energy is available, the quality of voltage supplied, the reliability in terms of the number of disruptions, the capacity in terms of power available, the consumption levels allowed per day, and affordability of the standard consumption package as a percentage of household income⁹. Energy metrics have been assessed quantitatively across several countries. Though some aspects, such as formal disconnections from energy service¹⁰, are translatable to the US context, these methods require normalizing many variables amongst different types of data and are overly broad for applications in areas where electricity access is relatively high and reliable.

Of such areas, the United Kingdom (UK) has a richer history of incorporating energy poverty formally into its government programs: since 2000, the UK has used some form of an energy burden metric to assess whether households are facing energy poverty and determine the level of support that they require as a result¹¹. This metric has evolved from a simple ratio of household income and energy expenditures to one that incorporates building efficiency ratings and average incomes in the community. While the European Union (EU) currently lacks a unified metric for energy poverty, similar metrics have been developed in member countries, such as a metric which compares incomes and expenditures to local averages and absolute heating needs to determine energy poverty levels in Italy¹² or a multidimensional index of building quality and ability to pay bills in Poland¹³.”

[Lines 219-227] “The slope of the N_h density lines in Figure 4.c indicates that these lower-income households seem to experience a different rate of return on N_h from the benefits of increased energy adoption than higher-income households. Why are certain households not receiving the same benefits of their fuel source across the distribution of incomes? This could be due to lower-income households’ low consumption, meaning that the potential savings from implementing energy efficiency measures are lower than for high-income households according to the prebound effect identified by Sunikka-Blank and Galvin³² and Cong et al.³³.”

Reviewer: For Figure 1, I am not seeing the main takeaway. What do we learn that is new to the literature? How do you define net income? I was surprised to see that most households had net income less than 60,000. For figure 1 it seems like it would be valuable to zoom in on the 0 – 100% It feels odd to see the x axis extend all the way to 400%. I understand that this is because those households may not have an annual income, but it still feels misleading. Since you use the circle sizes I wonder if it would be useful to have a circle that says > 4 million.

Response:

Figure 1 highlights the advantage of net energy return as a metric to evaluate energy burdens and energy poverty. This differs from studies that utilize an energy burden metric that limits the scale to 0-100%, thereby removing outliers that may actually be important to examine. We have clarified the explanation in the figure description and main text:

[Lines 216-139] “While E_b appears inversely correlated to net income, this is primarily driven by a long tail of households with zero or very low incomes, often with energy expenditures exceeding their incomes (approximately 180,000 households in the dataset have $E_b > 100\%$). Due to the structure of the E_b equation (Equation 1), the E_b of these households approaches infinity and cannot be captured on the standard 0-100% scale the metric is intended to be interpreted within: around 37,000 homes have an infinite energy burden. Since LEAD is estimated and provided for the express purpose of exploring low-income communities, we are hesitant to discard these households as outliers. N_h provides a framing of the same dataset that allows for exploration of most households on a similar scale without the long tail. N_h appears positively related to income, and most communities appear within a few orders of magnitude. Utilizing N_h avoids discarding low-income communities as outliers in energy poverty analysis. Furthermore, N_h offers a way to view the relationship between energy expenditures and income such that the wide disparity between those in broader poverty is immediately apparent: many households with moderate-to-high E_b s are actually higher-income households with high energy expenditures, making their N_h s quite high (e.g., >\$100 of income per \$1 of energy spending).”

Circle sizes are provided for reference only and do not indicate a continuous key. Only 1 circle representing greater than 4 million households is presented in Figure 1, so it seems to be a sensible maximum legend reference size.

Reviewer: Please move the AMI acronym definition to when the AMI is first mentioned.

Response:

We have reordered the definition of acronyms to their first introduction, including AMI.

Reviewer: What is equation 7 for? All equations should be referenced in the text and explained. Without this it leaves the reader wondering what the purpose of the information is.

Response:

We have combined the definition of the energy poverty line into one equation to alleviate this confusion, and referenced all equations in the text. The previous equation 7 was provided to define the energy poverty line used in the analysis, which is now outlined in Equation 3, described in the main text as follows:

[Lines 149-159] “While a variety of thresholds have been developed and explored, energy poor households in the US are commonly defined in terms of E_b as those with an expenditure of greater than 6% of household income on energy based on the logic that energy expenditures should not be greater than 20% of housing expenses, which themselves should not exceed 30% of household income⁸. Calibrating our N_h analysis to this level will help gauge different thresholds of energy poverty and benchmark the results of this paper to the energy poverty literature while acknowledging the continuum of experiences across household energy consumption. Translated into its relative level for N_h , the energy poverty line N_h^* is defined according to Equation 3 as approximately 16.

This means that a household that earns less than approximately \$16 of income for every dollar it spends on secondary energy will be considered to be in energy poverty by the traditional E_b accounting method. An N_h of approximately 16 or lower is equivalent to an E_b of 6% or higher.”

Reviewer: In general it is good practice to discuss the figure in the text before it is presented.

Response:

The figures are reordered to appear after their introduction and discussion and are introduced prior to appearing.

Reviewer: When talking about the households in figure 4 do you mean that 10% of census tracts experience energy poverty?

Response:

No, we have clarified this. The data refer to households, rather than census tracts. Census tracts are a category that households are categorized within in the representative sample from which

the data are extrapolated, like income brackets and primary heating fuel type. Therefore, the study does refer to the number of households in Figure 4 and throughout (e.g. 10% of households in the United States).

[Lines 401-406] “The dataset is assembled by applying an iterative proportional fitting (IPF) algorithm to cross-tabulations of household responses from the 2016 5-year American Community Survey (conducted by the US Census Bureau), which provides the samples for each cohort as Public Use Microdata Samples. IPF is a widely used spatial microsimulation method to allocate individuals (i.e., households) to zones (i.e., census tracts and utility service territories) while calibrating each zone’s characteristics to known quantities.”

[Lines 6-7] “Here, we develop a framework using Net Energy Analysis and socioeconomic data at the household level.”

[Lines 70-73] “The US Department of Energy (DOE) significantly improved upon Ross and Drehobl’s underlying methodology by assembling its Low-Income Energy Affordability Dataset (LEAD)²³, which estimates incomes and energy expenditures for most households in the US at a census tract scale.”

Reviewer: Also, the discussion about most households who are experiencing energy poverty also experiencing energy poverty seems misleading. Because energy poverty is a function of income it follows that the two would be correlated. Since they are not independent of each other it makes sense that if you are experiencing one then you are likely to experience the other.

Response:

We have added that the relationship between income and energy expenditures is non-linear and disproportionate in the beginning of the discussion section. While we agree that energy poverty is a good proxy for poverty, with 94% of households below the Federal Poverty Line also facing energy poverty, more than 5.2 million households above the Federal Poverty Line face this scarcity. More than 100,000 households have no incomes or energy expenditures greater than their incomes, and are excluded from a typical energy burden analysis. The purpose of this paper is to introduce a measure that can capture groups like these which would be excluded from other existing measures. Furthermore, N_i allows normalization of incomes on energy expenditure terms to make other comparisons and facilitate a discussion that is more focused on the resulting disparities observed, particularly compared with previous literature that focuses on simpler income-based metrics or limited efficiency-oriented estimates.

[Lines 278-286] “A sensible prior hypothesis is that everyone experiences the same efficiency from the energy system as measured by return on energy investment. Differences in absolute

outcomes may be related to the quantity of energy investment, but the marginal unit of energy consumed by one household should lead to as much benefit for that household as any other. However, here we see that $N_{h,s}$ are different among different groups of households in the US. This difference is often correlated to factors out of the households' control and even those related to persistent social inequalities, such as race and education. These striking disparities suggest the existence of deeply structural barriers to prosperity in the US. Energy is central to equity and economic prosperity, but the energy system appears to be regressive in that costs accrue disproportionately to those of lower-income levels.”

Reviewer: In the introduction, there was a lot of discussion about technology adoption, so I was surprised that this was not discussed in detail (with a figure) in the main text. Based on the introduction framing I experienced there to be some analysis on which communities are experiencing technology adoption barriers.

Response:

Much of this technology adoption discussion was removed from the introduction in favor of a more focused narrative on the energy poverty literature. We intend to follow this study that addresses these questions in future work.

[Lines 342-350] “Research using detailed qualitative sociological and public health interview data links energy burden with housing and energy policy⁴⁴. Other articles have linked technology adoption barriers to low-income communities of color^{45,46,47} and identified that households that are in the same peer social network often adopt technologies such as solar⁴⁸. While some studies already acknowledge the benefits of improving energy efficiency and equity outcomes through surveys and interviews⁴⁹, this study adds to the literature by building a comprehensive quantitative framework and empirical result based on a large dataset. This toolkit adds more quantitative backing to this body of qualitative work and can be used in future analysis with technology adoption data to identify strategic opportunities to improve energy efficiency and access to clean technology in a more equitable manner.”

Reviewer: I see in your methods you use the Rooftop Energy Potential of Low-Income Communities in America. I was not sure how this fed into your analysis. In the methods it would have been helpful to mention the equations this feeds into, and more detail about what it is used for in your main text analysis. Without this information it will be hard for others to replicate this work, and build on your analysis.

Response:

The REPLICA dataset provides some data types to categorize the net energy burden that are not available in LEAD such as the utility type, cost of electricity, racial composition, and education levels. All code and data used in this analysis will be available in an open-source format designed and documented for use by other researchers. Copies of the codebase have been included in the initial submission, with their public release delayed until after the peer-review process is complete. This feature is emphasized in the Code Availability and Data Availability sections. At this stage, we primarily use REPLICA for the demographic data it contains. We have clarified which data we use from REPLICA in the methods. The rooftop solar potential estimates in REPLICA will be foundational to future analysis:

[Lines 409-413] “The Renewable Energy Potential of Low-Income Communities in America (REPLICA) dataset includes the racial and education-level composition of each census tract used in the ultimate analysis⁵⁵. In addition to providing a simpler designation of cohorts for each census tract as described below in Methods-Treatment, REPLICA also includes estimates of the technical potential of rooftop solar and additional techno-economic variables (e.g. , demographics and electricity rates) that will be useful for future research.”

[Lines 463-466] “Characteristics such as the utility type and locale description are sourced from REPLICA and unavailable in LEAD, and we source the cost of electricity, racial composition, and education levels of census tracts from REPLICA for this analysis.”

[Lines 498-506] “**Code Availability**

The code and data to fully reproduce this paper are available on GitHub at **redacted for anonymity - copies of data and code have been submitted as part of peer-review** under the GNU Affero General Public License v3.0.

Data Availability

All data necessary for the composition of the source datasets used in this analysis are freely available from US government sources as open data. All functions to automatically retrieve and assemble these data and compiled versions of these data are made available to the user as part of the software referred to in Code Availability.”

[Lines 187-189] “By processing disparate data sources into a coherent structure and providing a convenient open-source tool for others to do the same, these data can be used in urban planning, public policy, and other relevant contexts.”

Reviewer: In general I think this paper covers an important topic (energy poverty, and who is experiencing that poverty), but I think the paper is lacking in the insights.

Response: Thank you - this study will provide more insights into how net energy metrics can be applied to better track and identify energy burdens across census tracts. In addition, net energy metrics identify justice concerns that are overlooked when using different energy burden metrics. The discussion now situates this paper within a body of energy burden literature.

Reviewer: One thing I would have liked to see is more detailed description of what value this paper adds to the multitude of studies that examine energy poverty in the US.

Response:

The contributions of this paper are more clearly stated specifically in the introduction and findings:

[Lines 104-105] “The main contribution of the study is the application of NER as an indicator of household energy poverty.”

The study adds to the qualitative survey and interview literature with a quantitative framework to provide high-quality large datasets to track the magnitude of energy poverty in the US and identify energy justice concerns.

[Lines 179-189] “Another contribution of this work is that by using N_h we can display these data spatially across the US to explore how different communities are experiencing energy outcomes as in Figure 2 and investigate specific communities at multiple scales such as census tract, county, state, and regional as in Figure 3. Furthermore, we can break down the data among meaningful subsets as in Figure 4 and examine state-by-state trends as in Figure 5. This is useful because the presence of energy inequity is harder to see in urban areas compared to rural areas for which census tracts are a larger physical area, or when certain household characteristics such as primary heating fuel are related to widely differing energy outcomes in the same area. Additionally, it shows a spatial variation of energy poverty that includes 5.3 million more households that would not be captured by traditional poverty metrics because their incomes are too low ($E_b > 100\%$) or too high (above the FPL). By processing disparate data sources into a coherent structure and providing a convenient open-source tool for others to do the same, these data can be used in urban planning, public policy, and other relevant contexts.”

Reviewer: Additionally, it seems that the paper was discussing households, but the data appears to be at the census tract level.

Response:

The data refer to households, rather than census tracts and are, therefore, the basis for a discussion about households. Census tracts are a category that households are categorized within in the representative sample from which the data are extrapolated, like income brackets and primary heating fuel type. Therefore, the study does refer to the number of households in Figure 4 and throughout (e.g. 10% of households in the United States).

[Lines 401-406] “The dataset is assembled by applying an iterative proportional fitting (IPF) algorithm to cross-tabulations of household responses from the 2016 5-year American Community Survey (conducted by the US Census Bureau), which provides the samples for each cohort as Public Use Microdata Samples. IPF is a widely used spatial microsimulation method to allocate individuals (i.e., households) to zones (i.e., census tracts and utility service territories) while calibrating each zone’s characteristics to known quantities.”

[Lines 6-7] “Here, we develop a framework using Net Energy Analysis and socioeconomic data at the household level.”

[Lines 70-73] “The US Department of Energy (DOE) significantly improved upon Ross and Drehobl’s underlying methodology by assembling its Low-Income Energy Affordability Dataset (LEAD)²³, which estimates incomes and energy expenditures for most households in the US at a census tract scale.”

Reviewer: Lastly, I think the methods needed more clarity. Some of the variables were defined twice, and some equations lacked explanations in the text which made the article hard to follow.

Response:

The methods have been revised to articulate each equation and variable used in the analysis, reduce the number of variables to four, and simplified the number of equations to three. The equations are also introduced in the main text with more explanation that would enable replication. Furthermore, all code and data used in this analysis will be available in an open-source format designed and documented for use by other researchers. Copies of the codebase have been included in the initial submission, with their public release delayed until after the peer-review process is complete. We have added emphasis on this feature throughout the study based on this feedback

[Lines 498-506] “**Code Availability**

The code and data to fully reproduce this paper are available on GitHub at **redacted for anonymity - copies of data and code have been submitted as part of peer-review** under the GNU Affero General Public License v3.0.

Data Availability

All data necessary for the composition of the source datasets used in this analysis are freely available from US government sources as open data. All functions to automatically retrieve and assemble these data and compiled versions of these data are made available to the user as part of the software referred to in Code Availability.”

[Lines 187-189] “By processing disparate data sources into a coherent structure and providing a convenient open-source tool for others to do the same, these data can be used in urban planning, public policy, and other relevant contexts.”

Reviewer: I hope my comments above are helpful to the authors as they continue down an important research path.

Response: Thanks for providing extremely valuable feedback that can help improve the contributions of this paper.

Reviewer #3 (Remarks to the Author):

Reviewer: This study has many promising and thought-provoking components, but it suffers from a bit of an “identity crisis.” As written, the manuscript jumps around and lacks a cohesive narrative, and I think attempts to do too much in a short amount of space. As a consequence, the potential contributions of the study are masked and the data and analysis presented is under-developed.

Response:

We made significant edits and rearrangements to the introduction and discussion sections to address these issues raised by the reviewers. Specifically, we focused the text on introducing, describing, and assessing the results of applying this new energy poverty metric to a nationwide dataset. We removed speculative statements and hypotheses that are better suited for a future analysis building on this work.

Reviewer: As an example of the confusing presentation, the introduction begins with a discussion of energy affordability, then moves into a micro-level discussion of limitations of the FPL before jumping back to macro-level discussion of the implications of energy affordability for social cohesion, climate change, and sustainable development, before providing a one paragraph summary of the empirical analysis presented in the paper followed by a general statement about facilitating a “sustainable and prosperous society built on equity and justice.” At this point as a reader, it is very difficult to know what the paper is about. Rewriting can certainly fix this, but the disjointed presentation points to a more general problem with the current manuscript which is that it is difficult to follow.

Response:

We reordered the introduction to address the context of previous work on energy equity and motivation for introducing a new metric into the academic conversation.

Reviewer: The main potential contribution of the study is the application of “net energy return” as an indicator of household energy poverty, but it is not fully-developed in the current manuscript. This is an intriguing idea, and the authors offer several reasons for why this measure has analytical advantages compared to the more traditional measure of energy burden. The paper would benefit from a more robust discussion here.

Response:

We incorporated this description of the potential contribution almost verbatim into the introduction and reorganized the text significantly to emphasize this contribution throughout.

Reviewer: In fact, the text in the caption to Figure 1 is quite enlightening, and I think can be moved to the main text to amplify the arguments. More generally, the authors should give more attention to this part of the manuscript. To make room given the limited word count, I would suggest the authors substantially cut back the discussion section, much of which is speculative and not supported by the analysis provided (more on this below).

Response:

Thanks for this suggestion. We moved the substantive portion of Figure 1 into the main text as recommended and expanded on these points and others like it, removing unsubstantiated discussion points as recommended below.

Reviewer: A second potential contribution of the paper is the analysis showing the spatial distribution of NER and its composition by socio-demographic characteristics. Here too, the manuscript would benefit from more detailed discussion of what Table 1 and Figures 2, 3, and 4 demonstrate. These graphics each present a large amount of data, but it is not obvious what a reader is supposed to take away from them.

Response:

We incorporated more direct descriptions of and references to the figures and the table to emphasize this point, as well as explicitly noting the advantages that these figures and the table demonstrate in the main text. In addition, we added parenthetical summary statistics to each section of the findings so that the reader can better understand the results of the analysis in context with the sentence describing the finding being discussed.

Reviewer: In addition, new metrics and data sources are introduced but not explained (e.g., Area Median Income, LEAD); the details are in the Methods section but the authors need to provide enough explanation in the main text so the reader can follow along.

Response:

We added brief explanations of these terms and sources when they are first introduced.

Reviewer: In addition to a more complete discussion of the main contributions of their study, the authors in my view need to be much more careful in their interpretation. The analysis contains a number of assertions that neither directly follow from the data presented nor are supported with appropriate citations. For instance, in the last full paragraph on p.4, the authors make several assertions about fossil fuel costs, environmental and health externalities, etc. without evidence.

Response:

The discussion has been edited to specifically focus on assertions from the data in this study. In other cases assertions have been removed or substantiated with a direct reference. With this revision, the discussion is more focused on the net energy metric results and the applicability of the tool presented to providing evidence-based policy analysis for US households, states, and regions.

Reviewer: The next partial paragraph similarly makes a statement about areas where inexpensive energy is available, and about housing quality and energy consumption, none of which come directly out of the data presented.

Response:

The discussion includes more explicit reference to the data presented in this analysis and other information that is not directly tied to the net energy return has been removed.

Reviewer: Later in this section, the authors discuss renters v. owners and households with solar power, but again this analysis does not follow directly from the data, or at least, not the way it is presented.

Response:

We have improved the association of claims with the relevant subfigures of Figure 4, and removed any remaining unsubstantiated phrases.

Reviewer: Similar problems emerge in the discussion section, where the authors speak of energy efficiency, intermittent grid access, utility disconnections, in-house combustion of biofuels and hydrocarbons, etc. This list goes on to include loans for solar programs, vertically integrated markets, the Roanoke Electric Membership Cooperation, and LIHEAP/WAP, etc.

Response:

We have reordered and rewritten the discussion to create a more focused narrative that connects directly to the findings, tying points to the data and literature where possible, and removing any unsubstantiated claims.

Reviewer: In sum, too much of the discussion is disconnected from the data and analysis presented, which is quite frustrating from a reader's standpoint. More importantly, I think the authors do a disservice to themselves because the scattered and speculative discussion takes away from an opportunity to better highlight the important contributions of the research. A broader discussion is certainly appropriate, but I might suggest a different approach in which the authors engage more deeply with other studies of household energy poverty, energy burden, and energy insecurity to explain what their measurement approach brings to the literature and to our understanding of these conditions.

Response:

In addition to adding more context about other studies of household energy poverty, energy burden, and energy insecurity in the introduction, we refer back to these studies in the conclusion to place our measurement approach in context. We also limited much of the previously wide-ranging discussion to a more focused discussion of how the measurement of energy poverty can help improve outcomes.

Reviewer: Some more specific comments

p.1, the problem characterized as a principal-agent dilemma is more commonly referred to as the split-incentive problem.

Response:

Thanks for identifying this, we now refer specifically to the split-incentive problem.

We have replaced references to "principal-agent dilemma" with "split incentive problem" and cited an appropriate source from the energy efficiency literature.

[Lines 238-243] "Renters face systemic disadvantages in the energy transition; they typically pay the home's energy costs while the landlord controls infrastructure upgrades, commonly understood as the split incentive problem³⁶. Tenure matters for more than just equity itself: renters are less likely to take actions to improve their N_hs due to a lack of property rights and split incentives. Even when action is taken to improve the energy efficiency of a rental building, tenants are less likely to see any economic benefits from it."

Reviewer: p.1, the assertion that “bureaucracy and institutional inertia” are holding back a rapid transition is vague and not substantiated.

Response:

This has been removed from the text.

Reviewer: p.2 the authors state that the study provides “a biophysical framework to evaluate the disparities among household net energy outcomes”, but this is not explained, and I frankly do not understand what the authors mean.

Response:

Net energy return is a metric that comes from biophysical and resource economics trying to understand the energy return on investment. We apply this net energy concept from biophysical and ecological economics to the study of energy burdens in households because that may be a relevant metric that can help identify disparities in household energy burdens that is not already being done in a systematic way.

We clarified this in the text by stating the meaning of biophysical rather than using it without context and cited a canonical text in biophysical economics regarding this description.

[Lines 100-111] “NEA offers potential support to understanding energy equity through the use of formally defined Energy Return Ratios (ERRs) that articulate the relationships among energy flows within complex systems²⁶. Net Energy Return (NER), which describes the newly released potential to do work as a result of some activity, is recommended as a basis for future analysis²⁷, especially in the study of macro-energy systems like the US residential housing stock²⁸. The main contribution of the study is the application of NER as an indicator of household energy poverty.

Here, we use LEAD to examine the relationship between energy spending and household income spatially for most households in the US at a census tract scale, with particular emphasis on how disparate household NERs signal economic disparities across socio-demographic characteristics such as race and ethnicity, income, and housing tenure. This empirical analysis fills a gap in the current discussion about energy equity by providing a framework to evaluate the disparities among household net energy outcomes that is aligned with the assessment of energy flows through biological, physical, and economic systems²⁹.”

Reviewer: p.3 The authors assert that energy poverty is commonly defined as a household spending 10% or more of its income on energy. The citation is to Bednar and Reames (2020), but

they are referring to fuel poverty measurement in the UK. In the US, I think 6% is more common, as in this report from ACEEE

(<https://www.aceee.org/sites/default/files/energy-affordability.pdf>).

Response:

Thanks for bringing this more common proportion of household spending to our attention and since this analysis is focused on the US, we have adjusted the threshold to 6% and referred to the ACEEE work, which has been pioneering in this area. These changes are implemented in the calculations and figures.

[Line 67] “Ross and Drehobl performed distinct urban¹⁶ and rural¹⁴ analyses to describe energy inequity in the US.”

[Lines 149-152] “While a variety of thresholds have been developed and explored, energy-poor households in the US are commonly defined in terms of E_b as those with an expenditure of greater than 6% of household income on energy based on the logic that energy expenditures should not be greater than 20% of housing expenses, which themselves should not exceed 30% of household income⁸.”

Reviewers' Comments:

Reviewer #1:

Remarks to the Author:

I want to thank the journal for the opportunity to review this revised manuscript.

I want to particularly thank the authors for taking up the concerns raised by the reviewers and addressing them so extensively.

I appreciate the care with which they undertook their revisions.

Overall, I found the revised manuscript much clearer and the argument clearer and stronger and ready to be shared with the scientific community.

Reviewer #2:

Remarks to the Author:

I believe the manuscript is stronger now that the authors have removed the technology adoption discussion and focused their narrative on the energy poverty literature.

The revision of the methods is also more clear and easier to follow. Well done on the review.

Reviewer #3:

Remarks to the Author:

The authors have done an exceptional job responding to my previous comments and suggestions. The revised manuscript provides much improved conceptual clarity and narrative cohesion, and is better organized in both explaining the utility of the proposed approach to measuring energy burden and demonstrating its important implications. The result is a really nice paper that makes a compelling case for the NEA framework.

My only minor suggestion is that the authors consider changing the title of the paper. The current title reads like a newspaper headline, and might instead suggest a title that clearly indicates that the paper is proposing a new measurement strategy to show disparities in energy burden.

REVIEWERS' COMMENTS

NCOMMS-21-25017A

Reviewer #1 (Remarks to the Author):

I want to thank the journal for the opportunity to review this revised manuscript.

I want to particularly thank the authors for taking up the concerns raised by the reviewers and addressing them so extensively.

I appreciate the care with which they undertook their revisions.

Overall, I found the revised manuscript much clearer and the argument clearer and stronger and ready to be shared with the scientific community.

Response: We thank Reviewer #1 for their positive comments and suggestions to strengthen the paper.

Reviewer #2 (Remarks to the Author):

I believe the manuscript is stronger now that the authors have removed the technology adoption discussion and focused their narrative on the energy poverty literature.

The revision of the methods is also more clear and easier to follow. Well done on the review.

Response: We thank Reviewer #2 for their constructive comments and suggestions to strengthen the paper.

Reviewer #3 (Remarks to the Author):

The authors have done an exceptional job responding to my previous comments and suggestions. The revised manuscript provides much improved conceptual clarity and narrative cohesion, and is better organized in both explaining the utility of the proposed approach to measuring energy burden and demonstrating its important implications. The result is a really nice paper that makes a compelling case for the NEA framework.

My only minor suggestion is that the authors consider changing the title of the paper. The current title reads like a newspaper headline, and might instead suggest a title that clearly indicates that the paper is proposing a new measurement strategy to show disparities in energy burden.

Response: We thank the reviewer for their insightful comments and suggestions. We have changed the title to reflect this as: "A new measurement strategy to identify disparities across household energy burdens"